# Potential for phenol biodegradation in cloud waters

Audrey Lallement[1], Ludovic Besaury[1], Elise Tixier[1], Martine Sancelme[1], Pierre Amato[1], Virginie Vinatier[1], Isabelle Canet[1], Olga V. Polyakova[2], Viatcheslay B. Artaev[3], Albert T. Lebedev[2], Laurent Deguillaume[4], Gilles Mailhot[1] and Anne-Marie Delort[1*]

[1]Université Clermont Auvergne, CNRS, SIGMA Clermont, Institut de Chimie de Clermont-Ferrand, F-63000 Clermont-Ferrand, France
[2]Lomonosov Moscow State University, Chemistry Department, Leninskie Gory 1/3, Moscow, 119991, Russia
[3]LECO Corporation, 3000 Lakeview Avenue, St. Joseph, Michigan, 49085, USA
[4]Université Clermont Auvergne, CNRS, Laboratoire de Météorologie Physique, F-63000 Clermont-Ferrand, France

*Correspondence to:* Anne-Marie Delort (a-marie.Delort@uca.fr)

**Abstract**. Phenol is toxic and can be found in many environments, in particular in the atmosphere due to its high volatility. It can be emitted directly from manufacturing processes or natural sources, and it can also result from benzene oxidation. Although phenol biodegradation by microorganisms has been studied in many environments, the cloud medium has not been investigated yet as the discovery of active microorganisms in cloud is rather recent.

The main objective of this work was to evaluate the potential degradation of phenol by cloud microorganisms. Phenol concentrations were measured by GC-MS on two cloud samples collected at the PUY station (summit of puy de Dôme, 1465 m a.s.l., France): they ranged from 0.15 to 0.21 µg L$^{-1}$.

The strategy for investigating its potential biodegradation involved a metatranscriptomic analysis and metabolic screening of bacterial strains from cloud water collected at the PUY station for phenol degradation capabilities (from the 145 tested strains, 33 were isolated for this work).

Among prokaryotic messenger RNA enriched metatranscriptomes obtained from 3 cloud water samples, different from those used for phenol quantification, we detected transcripts of genes coding for enzymes involved in phenol degradation (phenol monooxygenases and phenol hydroxylases) and its main degradation product, catechol (catechol 1,2-dioxygenases). These enzymes were likely from Gamma-proteobacteria, a dominant class in clouds, more specifically the genera *Acinetobacter* and *Pseudomonas*.

Bacterial isolates from cloud water samples (*Pseudomonas* spp., *Rhodococcus* spp. and strains from the Moraxellaceae family) were screened for their ability to degrade phenol: 93% of the 145 strains tested were positive. These findings highlight the possibility of phenol degradation by microorganisms in clouds.

**Main findings.** Metatranscriptomic analysis suggested that phenol could be biodegraded in-clouds, while 93% of 145 bacterial strains isolated from clouds were able to degrade phenol.

**Key words.** Cloud water**,** phenol, biodegradation, metatranscriptomics*,* puy de Dôme.

## 1 Introduction

Due to its toxicity, phenol is one of the main pollutants listed by U.S Environmental Protection Agency (US EPA list) and its concentration in drinking water is inspected and regulated (Michalowicz and Duda, 2007). In France, phenol limit concentration in drinking water is 0.5 µg L$^{-1}$. Phenol is issued from natural sources such as

organic matter decomposition and biomass burning (Schauer et al., 2001), but it mainly results from industrial processes. For instance phenol is involved in the production of oils, xylene, plastics, drugs, explosives, dyes, pesticides; it is also present in oil refining and wood and leather preservatives (Gami et al., 2014; Schummer et al., 2009). The annual phenol production exceeds 10.7 million tones worldwide in 2016 (Merchant Research & Consulting ltd). Phenol has an environmental impact, particularly on the aquatic biota (microorganisms, protozoa, invertebrates and vertebrates) (Babich and Davis, 1981; Duan et al., 2018). Phenol represents also a risk for human beings because it can be rapidly absorbed through the skin and by inhalation through the lungs. In particular it provokes cutaneous exfoliation and cardiac arrhythmias; it is also toxic to the liver and kidneys (Babich and Davis, 1981; Lober, 1987) (National Library of Medicine HSDB Database: https://toxnet.nlm.nih.gov/cgibin/sis/search/a?d-bs+hsdb :@term+@DOCNO+113).

Phenol can be found in all environmental compartments (soil, water), including the atmosphere (Atkinson et al., 1992; Rubio et al., 2012). Even if its volatility is low ($\leq$ 7% at 25 °C; The National Institute for Occupational Safety and Health (NIOSH)), phenol is present in the gas phase, but this polar compound can also be transferred to the aqueous phases of the atmosphere (rain, snow, clouds) thanks to its solubility described by the Henry's law constant (H = 3.2 $10^3$ M atm$^{-1}$ at 298 K and mass accommodation = 2.7 $10^{-2}$ at 283 K; (Harrison et al., 2002; Heal et al., 1995)). Phenol can also be formed by the oxidation of precursors such as benzene directly in the atmosphere both in the gas and the aqueous phase (Grosjean, 1991; Harrison et al., 2005; Herrmann et al., 2015; Vione et al., 2004). The production of phenol by the aqueous phase reactivity is expected to be less efficient than in the gas phase. Indeed, benzene is precursor of phenol but it will not accumulate in the droplet in significant amount due to its relatively low Henry's law constant (H = 1.8 $10^{-1}$ M atm$^{-1}$). Phenol concentration ranges from 2.8 to 8.9 µg L$^{-1}$ (0.03 to 0.09 µM) in cloud waters and it reaches up to 91.3 µg L$^{-1}$ (0,97 µM) in rain (Harrison et al., 2005; Schummer et al., 2009).

In the gas phase, phenol is transformed into nitrophenols either in the presence of HO$^{\bullet}$ and NO$_2^{\bullet}$ (during the day) or in the presence of NO$_3^{\bullet}$ and NO$_2^{\bullet}$ (during the night) (Atkinson et al., 1992; Olariu, 2001; Olariu et al., 2002). In the aqueous phase, phenol can undergo transformations that should be much faster than in the gas phase leading to the formation of nitrophenols (Vione et al., 2004). Recent studies show that direct photolysis should be competitive to the radical driven one for phenol (Rayne et al., 2009) and that phenol exposed to atmospherically relevant photochemical conditions lead to the production of low-volatile compounds such as light-absorbing molecules (HULIS). In-cloud processing of phenol can therefore be a source of Secondary Organic Aerosol (SOA) (Gilardoni et al., 2016; Sun et al., 2010).

A great number of studies have been conducted to assess the biodegradation of phenol by microorganisms including bacteria, fungi, yeast and algae in the context of environmental and water treatment chemistry (Michalowicz and Duda 2007). Most of those microorganisms were isolated from soils (including the rhizosphere) and waters (fresh and marine waters, waste waters and sediments) where contamination by phenol has been studied (Basha et al., 2010; Kafilzadeh et al., 2010; Michalowicz and Duda, 2007; Mishra and Kumar, 2017; Sandhu et al., 2009; Sridevi et al., 2012; Tian et al., 2017). Only one team has focused on atmospheric phenol uptake by microorganisms (Sandhu et al., 2007, 2009). They studied microbial community on leaves directly in contact with phenol in the air and found that they were able to degrade it. Many studies are based on direct measurement of the biodegradation activity of microbial isolates, in particular for biotechnological application in industrial effluent decontamination (Basha et al., 2010; Michalowicz and Duda, 2007; Mishra and

Kumar, 2017; Sridevi et al., 2012). Alternatively others used molecular based approaches and reported microbial
genes of phenol or catechol degrading enzymes (Brennerova et al., 2009; Fang et al., 2013; Sandhu et al., 2009;
Sharma et al., 2012; Silva et al., 2013; Suenaga et al., 2009) or gained knowledge from metatranscriptomic
analyses of microbial communities (Auffret et al., 2015). Ajaz et al. (2004) have identified thirty bacterial strains
resistant to phenol in garden soil and Padmanabhan et al. (2003) have done DNA-SIP with $^{13}$C-labeled phenol to
identify 6 phenol-degrading populations in soil thanks to 16S rRNA gene analysis. Main bacterial genera able to
biodegrade phenol are *Pseudomonas*, *Rhodococcus*, *Acinetobacter* and *Bacillus*, other genera are also described
such as *Arthrobacter*, *Alcalinogenes, Burkholderia, Thauera*, *etc.* (Basha et al., 2010; Fang et al., 2013; Jadeja et
al., 2014; Michalowicz and Duda, 2007; Padmanabhan et al., 2003; Silva et al., 2013). Major biodegradation
pathways for aerobic bacteria have been established (Figure 1) : first phenol can be oxidized into catechol by
phenol hydroxylases or phenol monooxygenases, then the ring cleavage can be catalyzed by dioxygenases,
catechol 1,2-dioxygenase produces cis-cis-muconate ("ortho" pathway) while catechol 2,3-dioxygenase leads to
2-hydroxymuconate semialdehyde ("meta" pathway). Finally these products are integrated in the central
metabolism of the bacteria and end up in $CO_2$ production (Basha et al., 2010). Alternative pathways have been
described with anaerobic microorganisms. In these cases, phenol is carboxylated by a carboxylase in the para
position to produce 4-hydroxybenzoate and this metabolite is further metabolized in benzoyl-CoA *via* anaerobic
routes before its ring opening step (Basha et al., 2010).
Although phenol is present in clouds, to our knowledge its transformation by microorganisms in these specific
environments has never been assessed. Bacterial density usually ranges from $10^4$ to $10^5$ cells per mL of cloud
water (Vaïtilingom et al., 2012). In spite of the numerous atmospheric stresses, it has been shown that
microorganisms can survive in clouds, maintain metabolic activity and degrade organic compounds (Delort et
al., 2010, 2017; Hill et al., 2007; Sattler et al., 2001; Vaïtilingom et al., 2013). Among bacteria known for phenol
degradation, *Pseudomonas* (Gamma-proteobacteria) and *Rhodococcus* (Actinobacteria) notably are frequently
found viable and potentially active in clouds (Amato et al., 2017a, 2017b).
The aim of this work was to explore the potential for phenol biodegradation in clouds. First, phenol
concentration was quantified in atmospheric waters, and cloud water metatranscriptomes were checked for the
presence of transcripts of phenol-degrading genes; second, bacterial strains isolated from cloud water were
screened for phenol biodegradation ability.
**2 Materials and Methods**
**2.1 Chemical reagents**
Phenol (>99%) and hydrogen peroxide (30%) were obtained from Fluka, sodium chloride (>99%),
dichloromethane (>99.8%) and sulfuric acid (>95-97%) were from Sigma Aldrich, acetonitrile (>99.9%) was
from VWR Chemicals, NaOH (99%) from Merck, and $MgSO_4$ (>98%) from Carlo Erba reactifs – SDS.
**2.2 Cloud water analysis**
**Cloud sampling:** Cloud waters have been sampled at the PUY station (summit of the puy de Dôme, 1 465 m
above the sea level, 45°46' North, 2°57' East, France) which is part of the atmospheric survey networks EMEP
(the European Monitoring and Evaluation Programme), GAW (Global Atmosphere Watch), and ACTRIS
(Aerosols, Clouds, and Trace gases Research Infrastructure). The sampling site is fully described in Deguillaume
et al. (2014). The global meteorological context was examined through 120 h back-trajectories of the air masses
sampled using the HYSPLIT model (HYbrid Single-Particle Lagrangian Inte-grated Trajectory). Two cloud
water samples collected in 2016 (October 21[th] and October 26[th]) were analyzed in this work for phenol
quantification by GC-MS. Three other samples were previously sampled and analyzed in 2013 (November 05[th]),
2014 (June 27[th]) and 2016 (February 16[th]) (Lebedev et al., 2018). Samples were collected using a sterilized cloud
droplet impactor and immediately filtered through Minisart® PES filter (0.22 µm porosity; Sartorius, Germany)
under sterilized conditions; these have been stored at -25 °C.
**GC-MS analysis:** Sample preparation was carried out according to US EPA 8270 method. Prior utilization, all
the glassware was cleaned with piranha reagent composed of 6 mL of sulfuric acid mixed with 2 mL of hydrogen
peroxide. The reagents were kept in the glassware for one night and after all the glasses were washed two times
with ultrapure water and two times with dichloromethane. With clean dishes, cloud waters kept frozen were
melted at room temperature and the pH adjusted to pH = 2 and 11. Organic compounds were extracted tree times
with dichloromethane (keeping the ratio 10 mL of water for 1 mL of $CH_2Cl_2$). All the dichloromethane fractions
were then dried with $MgSO_4$ and evaporated to 1 mL using a rotary evaporator under reduced pressure;
temperature of the water bath was 20 °C. Samples were kept at 4 °C until analysis
All analyses related to cloud samples collected the 21[th] and 26[th] of October 2016 were performed at Saint Joseph,
MI at LECO Corporation (USA). Accurate GC-MS measurements were performed with a high resolution time-
of-flight mass spectrometer Pegasus® GC-HRT in GC mode (software ChromaTOF-HRT). The obtained EI
mass spectra were used for phenol identification by utilizing high mass accuracy data and retention time
(Lebedev et al., 2013). Phenol concentrations were measured using naphthalene D8 as internal standard.
Response factor (0.7) was calculated using standard solution of phenol. Phenol concentrations measured in cloud
water samples collected the 5-11-2013, 27-06-2014 and 16-02-2016 are extracted from Lebedev et al. (2018).
**2.3 Analysis of metatranscriptomes**
**Transcriptomic analysis:** Cloud water samples were collected on November 17[th], 2014, for three consecutive
periods of 5 h. The cloud air mass origin remained stable over the duration of sampling as attested by air mass
backward trajectories (Figure SM1). The clouds droplets collected by impaction were immediately transferred by
gravity into sterile collection bottles (Nalgene, Rochester, U.S.A) through sterile (autoclaved) silicone tubing.
Before cloud sampling operations started, control samples were made by pouring 200 mL of sterile water into
the collection device and through the tubing, and by processing it in parallel of the cloud water samples,
including sequencing and data treatment. These controls were clearly distinct from samples: based on their
contribution to identified ribosome sequences, these contained mostly Enterobacteriaceae (66%), Dikarya
(9.2%), Streptococcaceae (5.4%), Vibrionaceae (2.8%) and Micrococcaceae (1.2%), *i.e.* not the taxa of interest
here. Conservatively, the sequences present in controls were further removed from sample files (BWA-MEM; li
et al., 2013). Immediately after collection, water sample were filtered (MoBio 14880-50-WF) within an UV-
sterilized laminar flow hood installed at the sampling site. The filters were then put into ~5 mL of RNA Later
solution (Sigma, Steiheim, Germany) and stored at -80 °C until further processing. Briefly, total RNA were
extracted from filter halves using MoBio Power Water RNA kit and bacterial ribosomal RNA were depleted
using MICROB*Express* Bacterial mRNA Enrichment kit (Life Technologies). Metatranscriptomes of the
messenger RNAs were then obtained by multiple displacement amplification using REPLI-g WGA & WTA kit.;
Shotgun libraries were sequenced on Illumina MiSeq paired-end 2*300 bp. Sequencing reads were quality
checked (FastQC, Andrews, 2010) and trimmed (PRINSEQ-Lite, Schmieder and Edwards, 2011) before
assembling the mate pairs using PANDA-SEQ (Masella et al., 2012). Annotations were made against
UNIPROTKB database (Leinonen et al., 2006), including protein sequences for bacteria, archaea, and fungi
using BLASTX software (best hits with e-value $< 10^{-4}$). All steps were performed using custom scripts. The
sequence files have been deposited to European Nucleotide Archive (ENA) under the study accession number
PRJEB25802.
**Bioinformatics treatment:** Known enzymes involved in phenol degradation were found in KEGG Database
(see Figure 1). We only focused on aerobic metabolism as cloud environment is highly oxidative. Four
nucleotide sequences databases were created from NCBI corresponding to phenol hydroxylases (69 sequences),
phenol monooxygenases (29 sequences); catechol (regrouping catechol 1,2-dioxygenases and catechol 2,3-
dioxygenases) (145 sequences) and a fourth database including genes coding for putative phenol degradation
enzymes (38 sequences). The sequences from the cloud metatranscriptomes corresponding to the different
created databases were then extracted suing Bowtie2 (very-sensitive option; Langmead and Salzberg, 2012). The
affiliation of the extracted sequences was determined using blastn on a local server (e-value = 0.00001; Camacho
et al., 2009).
**2.4 Biodegradation of phenol by bacterial strains from cloud waters**
**Bacterial strains**: Bacterial strains were isolated from cloud waters sampled at the PUY station and identified as
previously described in Vaïtilingom et al. (2012). From our lab strain collection, we choose all the potential
bacteria that could biodegrade phenol. From the 145 strains tested, 33 of the strains were isolated for this work,
the others were published earlier (see Table SM1).
119 *Pseudomonas*, 24 *Rhodococcus* strains and 2 strains form the Moraxella family were selected for the
screening of phenol degradation (see Table SM1). More precisely Pseudomonas strains included 4 *P.*
*fluorescens*, 10 *P. graminis,* 1 *P. grimontii*, 2 *P. poae*, 1 *P. reactans*, 1 *P. reinekii*, 3 *P. rhizosphaerae*, 35 *P.*
*syringae*, 2 *P. trivialis*, 2 *P. veronii*, 1 *P. viridiflava* and 57 *Pseudomonas sp. Rhodococcus* strains included 1 *R.*
*erythropolis*, 1 *R. enclensis* and 22 *Rhodococcus sp.* Moraxella family strains were 1 *Moraxella sp* and 1
*Psychrobacter sp.*
*Pseudomonas* and *Rhodococcus* strains represent 20.4% and 4.10% of the 584 strains of our cloud bacterial
collection. From our experience, at the genus level, *Pseudomonas* and *Rhodococcus* are among the most frequent
bacteria in clouds: *Pseudomonas* strains in particular have been frequently isolated by culture (Vaïtilingom et al.,
2012; Joly et al., 2013), and both targeted and untargeted molecular analyses (and metagenomes, respectively)
demonstrated high occurrence in the bacterial communities. These represented 0.1 to >2% of the prokaryotes
ribosome sequences in amplicon sequencing investigations (Amato et al., 2017a). Based on the biomass in
clouds (~$10^4$ bacteria cells mL$^{-1}$; Vaïtilingom et al., 2012), and assuming even ribosome amplification between
bacterial groups, we can infer the presence of ~$10^3$ *Pseudomonas* mL$^{-1}$ and ~$10^2$ *Rhodococcus* mL$^{-1}$ of cloud
water.
**Incubations**: All the strains were grown in 25 ml of R2A medium for 48 h at 17 °C, 130 rpm (Reasoner and
Geldreich, 1985). Then cultures were centrifuged at 4 000 rpm for 15 min. Bacteria pellets were rinsed first with
5 mL of NaCl 0.8% and after with Volvic® mineral water previously sterilized by filtration under sterile
conditions using a 0.22 µm PES filter. Cells were re-suspended in 5 mL of 0.1 mM phenol solution, prepared in
Volvic® mineral water, and incubated at 17 °C, 130 rpm agitation during 5 days in the dark. To know the
concentration, the OD for each strain was taken during the experiment. Strains concentration was around $10^9$
cells mL$^{-1}$. The ratio number of bacterial cells / phenol concentration was kept to that measured in cloud waters,
in clear all the concentrations were multiplied by a factor $10^4$. Indeed the mean bacteria concentration is around
$10^5$ cells mL$^{-1}$ of cloud water while that of phenol can reach 0.008 µM (0.74 µg L$^{-1}$, see the result section) in
clouds collected at the PUY station. We showed in the past that when the cell / substrate ratios are kept constant
the rates of biodegradation are constant (Vaïtilingom et al., 2010). The temperature (17 °C) corresponds to the
average temperature at the PUY station in summer under cloud condition. It is well known that under culture
conditions in a laboratory, a lag time can be observed before bacteria starts to biodegrade phenol that
corresponds to the induction period of the gene expression (Al-Khalid and El-Naas, 2012).
Before sampling, evaporation of water has been compensated by adding Volvic® mineral water. A control
experiment was performed by incubating phenol without bacteria; phenol concentration remained stable with
time (0.1 mM of phenol was obtained at the end of the experiment). For phenol quantification over time in the
incubation experiments, 600 µL samples were centrifuged at 12 500 rpm for 3 min and the supernatants were
kept frozen until HPLC analysis.
**Phenol HPLC analysis:** Before analysis, all the samples were filtered on H-PTFE filter (pore size at 0.2 µm and
diameter of 13 mm from Macherey-nagel, Germany). Phenol detection was done on HPLC VWR Hitachi
Chromaster apparatus fitted with a DAD detector and driven by Chromaster software. Isocratic mode was used
with a reverse phase endcapped column (LiChrospher® RP-18, 150 mm x 4.6 mm, 5 µm, 100 Å). Mobile phase
was composed of acetonitrile and filtered water (Durapore® membrane filters, 0.45 µm HVLP type, Ireland) in
25/75 ratio with a flow rate at 1.2 mL min$^{-1}$ (adapted from Zhai, 2012). Sample injection volume was 50 µL,
spectra were recorded at 272 nm and the run time was 10 min.
**Phenol degradation:** Percentage of phenol degradation was calculated by the following equation:
$$\text{Phenol degradation } (\%) = 100 - \frac{[Phenol]_{final} \times 100}{[Phenol]_{initial}} \quad (1)$$

Limit of phenol quantification was 3.8 µM. Strains are not considered active below 5% of phenol degradation,
corresponding to 5 µM.
Comparison of strain phenol degradation abilities was done using a non-parametrical Kruskal-Wallis test (p.
value < 0.05) with Past software.
**3 Results**
**3.1 Phenol quantification in cloud waters**
The objective of this paper was to explore the ability of microorganisms isolated or present in cloud waters
collected at the PUY station to degrade phenol. From the 145 tested strains, 33 were isolated in this work, the
others were already published (Table SM1). We first checked its presence in cloud waters sampled at the PUY
station by performing GC-MS analysis. Figure SM2 presents the back trajectories of the air masses
corresponding to the 5 cloud events at the PUY station. The air mass origins of the 5 cloud samples determined
from these back trajectories were classified as described in Deguillaume et al. (2014) and are reported in Table 1.
The GC-MS analysis performed on cloud samples allowed reliable identification and quantification of phenol in
all samples (Table 1), the measured phenol concentrations ranged from 0.15 to 0.74 µg L$^{-1}$. Figure SM3 (A)
represents, as a typical example, the total ion current chromatogram of the cloud sample collected the 16[th] of
February 2016, the corresponding mass-chromatogram based on the ion 94 current (characteristic for phenol) is
represented in Figure SM3 (B). Quantification was done using similar mass chromatograms of all samples and
the identification was proven by the correct retention time and accurate mass measurements (calculated:
94.0413; experimental: 94.0414).
**3.2 Possibility of in-cloud phenol degradation by the cloud microbiome using a meta-transcriptomic**
**analysis**
The presence of transcripts involved in the biodegradation of phenol (Figure 1) was investigated from
prokaryotic messenger RNA enriched metatranscriptomes obtained from 3 cloud water samples. Sequence data
were looked for the presence of transcripts of genes involved in phenol biodegradation among the 281 sequences
included in our database (more details about the affiliation of the sequences are given in Table SM2).
Gene transcripts were detected for all the enzymes, except the catechol 2,3-dioxygenase, showing a possible
implication of the microorganisms in the degradation of phenol in cloud (Figure 2). However the number of hits
and the relative abundance of the transcripts coding for the different enzymes varied according the cloud
samples. Two hundred fifty-seven hits (sequence homology) could be counted for cloud 2, for only 70 in cloud 1
and 130 in cloud 3. Transcripts corresponding to the enzyme involved in the first step of oxidation of phenol
leading to catechols (phenol hydroxylases and phenol monooxygenases) were the most abundant in clouds 2 and
3 while those corresponding to the cleavage of the catechol ring (catechol 1,2-dioxygenase) were dominant in
cloud 1. For all the samples the transcripts corresponding to putative phenol degradation enzyme pathways (i.e.
none explicitly described enzymes) remained low. However the slight differences observed between the 3 cloud
samples are not significant when analyzed by a non-parametrical Kruskal-Wallis test.
Figure 3 presents the relative abundance of putative taxonomic affiliation of microorganisms involved in phenol
biodegradation, based on the information associated with sequences in the databases. All the sequences were
affiliated with Gamma-proteobacteria, from only two genera, namely *Pseudomonas* and *Acinetobacter*,
corresponding to only four species (*P. fluorescens, A. gyllenbergii, A. oleivorans* and *A. pitii*) matched with
cloud transcripts, among a total of 50 (Table SM2). This very low diversity was unexpected considering that
sequences from 50 bacterial genera including 109 species were used for our search in data bases. In addition the
relative abundance of sequences affiliated to a bacterial species varied a lot with the considered enzymes and
clouds (Figure 3).
Gamma-proteobacteria were found to contribute up to 21% of the ribosome sequences identified in bacteria in
targeted sequencing investigations. *Pseudomonas* in particular was highlighted as one of the most represented
genus (contributing alone up to 2% of the ribosome sequences) and most active genera based on its
representation in transcriptomes and consecutive high ribosomal cDNA:DNA ratio (Amato et al, 2017a; Figure
SM4). *Acinetobacter* and *Rhodococcus* were much less represented (<0.1% of the ribosome sequences) but also
accounted for groups of interest regarding potential metabolic activity.
*Rhodococcus* were previously isolated from clouds at the PUY station (Vaïtilingom et al., 2012) but genes for
phenol degradation affiliated with this genus were not detected here.

**3.3 Screening of bacterial strains isolated from cloud waters for their ability to biodegrade phenol**

From our strain collection of 826 culturable microorganisms isolated from clouds collected at the PUY station
between March 2003 and June 2016, we selected strains belonging to genera of interest concerning their
potential ability for phenol biodegradation. We choose to test specifically *Pseudomonas* and *Acinteobacter*
strains as they were detected in our metatranscriptomic analysis. As no *Acinetobacter* was available in our
bacterial collection we choose closely related genera namely two strains of *Moraxella* and *Psychrobacter*. In
addition *Rhodococcus* is well-known to biodegrade phenol in the literature (as well as *Pseudomonas* and
*Acinetobacter*). *Pseudomonas* and *Rhodococcus* are also the most frequently found genera in culturable bacteria
from clouds (Renard et al., 2016; Vaïtilingom et al., 2012). Altogether 145 bacterial strains were tested (Table
SM1). The percentage of phenol degradation measured by HPLC after 5 days of incubation at 17 °C is reported
in Figure 4 and in Table SM1. As the objective of the work was to perform a large screening with different types
of cells, incubation duration of 5 days was chosen to be sure that the induction period necessary for laboratory
experiments was long enough to be able to detect biodegradation ability for all the tested cells. This time is quite
long for a cloud but the objective here is not to evaluate a rate of biodegradation but to investigate the potential
of biodegradation of microorganisms present in cloud waters.
We found that 93.1% of the 145 tested strains were able to degrade phenol after 5 days of incubation. Globally,
in our experimental conditions, all the families tested were very good phenol degraders (see Figure 4A). No
significant difference was found in the capacity of phenol degradation between *Pseudomonas*, *Rhodococcus* and
*Moraxellaceae* strains. A focus on the *Pseudomonas* strains according to their species is presented in Figure 4B.
The mean capacity of phenol degradation varied between 31 and 67% (for *Pseudomonas rhizosphaerae* and
*Pseudomonas graminis* respectively), however no significant difference was observed between the species
according to the Kruskal-Wallis test. Considering specifically *Pseudomonas syringae* strains which are the most
abundant species among cultural strains present in cloud waters (Renard et al., 2016), only 2 of them out of 35
were not capable of degrading phenol (strains PDD-32b-31and PDD-69b-20, see Table SM1).

**4 Discussion and conclusion**

Phenol was present in the cloud water samples at concentrations ranging from 0.15 to 0.74 µg L$^{-1}$; these values
are within the range usually measured in atmospheric waters at remote sampling sites (3.0 to 5.4 µg L$^{-1}$, Harrison
et al., 2005), but globally the concentrations measured at the PUY station are rather in the lower range of values.
Although the concentration of phenol remains within the same order of magnitude in the 5 cloud samples, it
seems that the origin of the air masses had an impact on this concentration; it was 3 times lower in non-polluted
air masses (West) than in polluted ones (North West/North).
The results reported combining molecular approach and biodegradation assays involving culturable bacteria
indicate that phenol-degrading microorganisms are present in clouds. Molecular approach allowed detection of
transcripts belonging to *Pseudomonas* and *Acinetobacter* strains but not for the sequences of the other strains
present in Table SM2. It was surprising not to find *Rhodococcus* sequences as this genus is well-known to
degrade phenol as reported in the literature. In parallel *Rhodococcus* strains isolated from clouds were very

active phenol degraders but no *Acinetobacter* have been isolated from clouds. This difference reflects the complementarity but also the bias of each approach (molecular *vs* cultural). Meta-transcriptomic can be biased by technical issues (extraction, sequencing, *etc.*) or by the creation of uncomplete database. In the future, the database for phenol degradation could be improved by integrating more sequences, especially considering other data banks than NCBI. For instance the catechol operon sequences of *Pseudomonas synringae* (Berge et al., 2014) could be added to the data base. We recently published the genome sequence of *Pseudomonas syringae* 32b-74, *Pseudomonas graminis* 13b-3 and *Rhodococcus enclensis* 23b-28 which are degrading phenol (Table SM1) (Besaury et al., 2017a, b; Lallement et al., 2017); they could be used to implement the database. Finally in the future the genome of many phenol degraders (Table SM1) could be also sequenced and integrated.

On the other hand it is well known that culturable microorganisms only represent 1% or less than the total community, notably in clouds (Amann et al., 1995, Vaïtilingom et al., 2012). Strains of *Acinetobacter*, *Pseudomonas* and *Rhodococcus* genera are known to degrade phenol in other environments (Basha et al., 2010; Gami et al., 2014; Michalowicz and Duda, 2007; Sandhu et al., 2007). The cloud microbiota as described from a culturable approach harbors species usually affiliated with the phyllosphere (Amato et al., 2017b; Vaïtilingom et al., 2012). Sandhu et al. (2007) explored the presence of phenol degraders among microbial communities on plant leaves. They did not find *Pseudomonas* but they isolated *Acinetobacter* and *Rhodococcus* strains, and noticed globally a low diversity of phenol degraders. Only the genes encoding for the ortho pathway for phenol degradation that involves the catechol 1,2-dioxygenase activity were present in both Proteobacteria and Actinobacteria. Similarly, we did not find transcripts of genes coding for catechol 2,3-dioxygenase but only those coding for phenol hydroxylase, phenol monooxygenease and catechol 1,2-dioxygenase. In principle bacteria can have either *ortho* or *meta* pathways, or both, but their expression is dependent on phenol concentration. The enzyme catechol 1,2-dioxygenase is produced at low phenol concentration while catechol 2,3-dioxygenase enzymes become dominant at high phenol concentrations (3 mM) (Sandhu et al., 2009). This might explain why bacteria from clouds and the phyllosphere only produce catechol 1,2-dioxygenase as the phenol concentration in the atmosphere is much lower than in polluted surface water for instance (in the range of a few µg L$^{-1}$ versus 100 to 1000 µg L$^{-1}$) (Gami et al., 2014; Harrison et al., 2005; Schummer et al., 2009; Sturaro et al., 2010).

In our study we focused on *Pseudomonas* strains as they are the most frequent culturable strains (Vaïtilingom et al., 2012) and belong to the most active strains in cloud waters (Amato et al., 2017a). We observed that these strains likely issued from the phyllosphere, *P. graminis P. syringae, P. fluorescence, P. poae* and *P. viriflava* were able to degrade phenol. In the literature, *P. aeruginosa* and *P. putida* are the most popular phenol degraders (Basha et al., 2010; Der Yang and Humphrey, 1975; Erhan et al., 2004; Gami et al., 2014; Kumar et al., 2005; Molin and Nilsson, 1985). Interestingly Bartoli et al. (2015) showed that the genome of several *P. syringae* pathogens of woody plants contained a catechol operon, while it was not the case for other *P. syringae* strains pathogens of herbaceous plants. These results strongly suggested that the presence of enzymes present in the catechol pathway could help the degradation of aromatics present in lignins. In addition Berge et al. (2014) showed that some *P. syringae* strains from phylogroups 1 and 3 that were Ice Nuclei Active (INA[+]) also contained the catechol operon. In our case we also measured the ice nucleation activity of the 35 *Pseudomonas syringae* strains as described in Joly et al. (2013). Figure SM5 presents the strains which were INA[+] (T>-8 °C) *versus* their phenol degradation ability. Among the phenol degrader, 57.6% of the bacteria were INA[+].

Clouds can be considered as medium for microorganism transport and INA$^+$ bacteria are suspected to induce precipitations and thus participate to the water cycle (Morris et al., 2008). Consequently, the presence of *Pseudomonas syringae* in clouds combining ice nucleation and phenol degradation properties can be of major importance for the pathogenicity on woody plants, in terms of epidemiology, dispersion of pathogens and emergence of plant diseases.

We showed that microorganisms from clouds were able to degrade phenol. The question raises what is the potential impact of this biotransformation on the fate of phenol in real clouds? First the presence of transcripts of phenol-degrading enzymes measured directly *in situ* demonstrates a real in-cloud activity of microorganisms. However, these data do not give any exact quantitative contribution of the microbial activity to the phenol transformation in real clouds. On the other hand the large screening performed with selected cloud strains showed that they have the enzymatic equipment for phenol degradation. Future work should be conducted to evaluate this potential for phenol biodegradation in real clouds where a larger microbial diversity is present. In particular, precise biodegradation rates should be determined under "realistic cloud conditions" to evaluate its real impact. It will be also very important to compare the relative contribution of biological degradation *versus* radical chemistry, especially with photochemistry. It is well known that phenol can react with $^\bullet$OH, NO$_2$$^\bullet$, NO$_3$$^\bullet$ radicals alone or in combination to give rise to catechol, 2-nitrophenol, 4-nitrophenol and 2,4-diniropenol, these compounds can be further degraded in intermediates after the ring cleavage (Harrison et al., 2005).

In addition of examining the presence of phenol degrading pathways, we also looked for biological pathways leading to the potential formation of phenol from benzene (Choi et al., 2013; Tao et al., 2004), involving toluene monooxygenases in 8 species of Actinobacteria, Alpha- and Beta-proteobacteria (Table SM3). None of these sequences were found in the cloud prokaryote metatranscriptomes. This result should be confirmed by incubating cloud microorganisms directly with benzene to assess the real potential of cloud microorganism to produce phenol under these conditions. Consequently the origin of phenol in cloud waters could only result of the mass transfer from the gas phase to the aqueous phase or of the production *via* radical processes in the atmosphere. For instance the production of phenol in the gas phase can result from the reactivity of $^\bullet$OH radical with benzene (Grosjean, 1991; Volkamer et al., 2002). Considering that benzene has a very low solubility in water, it is likely that the production of phenol mainly occurs in the gas phase and is transferred to the water phase. The contributions of biotic or abiotic transformation of benzene into phenol in the water phase should remain minor processes.

In conclusion, this is the first report showing that cloud water is inhabited by microorganisms that have phenol degradation ability. The study was centered on bacteria present in cloud waters collected at the PUY station where phenol concentrations were measured by GC-MS and found in the range of 0.15 to 0.74 µg L$^{-1}$. Metatranscriptomic analysis suggested that phenol could be biodegraded in-clouds, while a large screening of isolated strains showed that the enzymatic equipment to degrade phenol was not rare. These two combined approaches suggested that *Pseudomonas*, *Acinetobacter* and *Rhodococcus* strains were the major genera potentially involved in phenol biodegradation. Further work is needed to evaluate the relative contribution of this biological activity and radical chemistry (particularly photochemistry) to phenol transformation. For that, experiments will be set up to measure phenol biodegradation rates under realistic cloud conditions and compare them with abiotic degradation rates. This will bring valuable information to better describe the fate of this pollutant in the atmosphere. Since phenol is highly toxic and is one of the main pollutants listed by U.S

Environmental Protection Agency (US EPA, 1981), this work will help to better assess its impact on health and air quality. The most probably, microorganisms could participate to a natural remediation process of the atmosphere.

**Acknowledgments**

This work was mainly funded by the French ANR program BIOCAP (ANR-13-BS06-0004), the ANR-DFG program CHLOROFILTER (ANR-DFG-14-CE35-005-02) and CNRS EC2CO program FONCOMIC.

The authors also acknowledge the financial support from the Regional Council of Auvergne, from the Observatoire de Physique du Globe de Clermont-Ferrand (OPGC), from the Fédération de Recherche en Environnement through the CPER Environnement founded by Région Auvergne-Rhône-Alpes, the French ministry, and FEDER from the European community

**Ethics statements**

This work does not involve human or animal subject. There is no ethical problem.

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

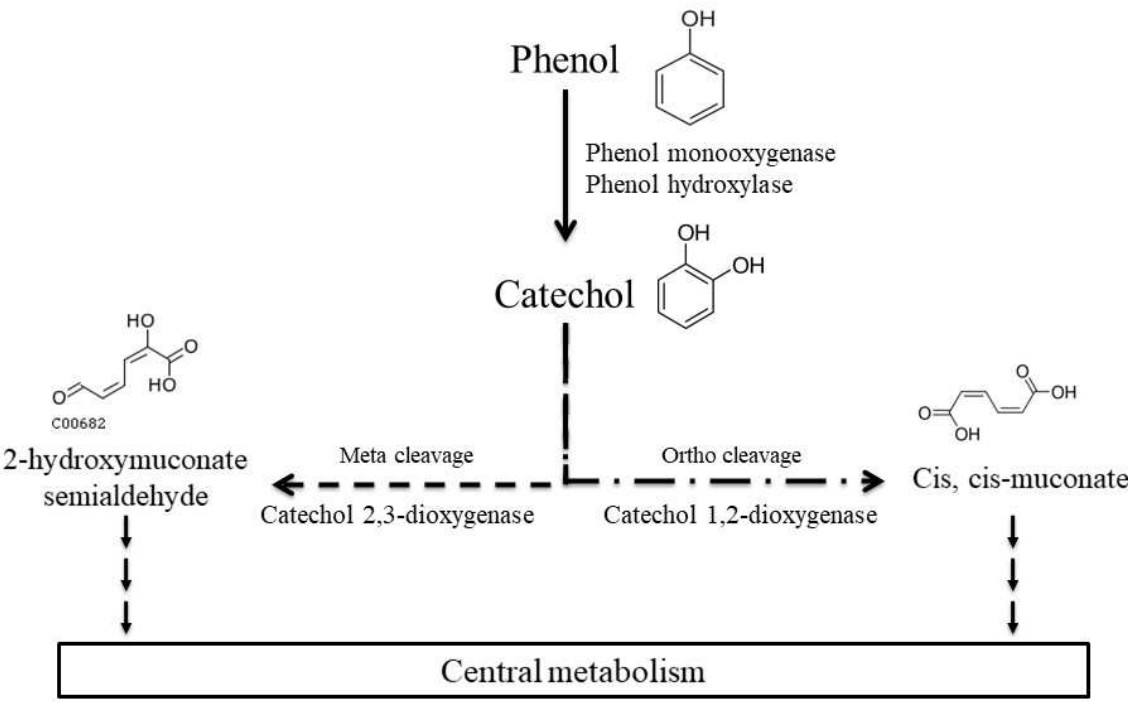


**Figure 1: Main phenol biodegradation pathways described for aerobic microorganisms as referred in KEGG database.**

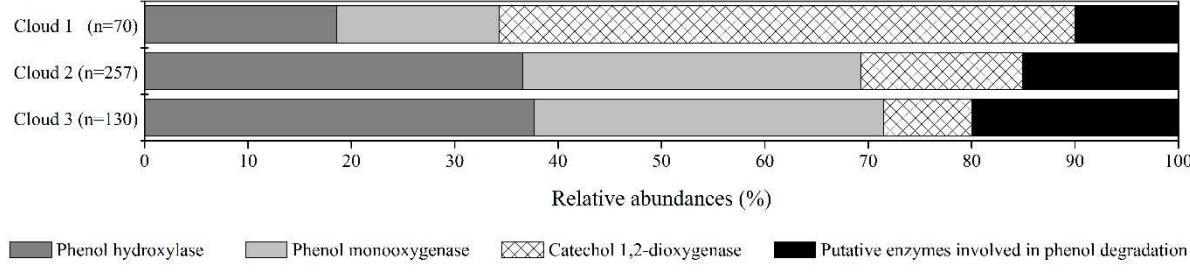


**Figure 2: Relative abundance of transcripts in cloud waters encoding for enzymes involved in phenol degradation**
**pathways. The absolute total number of hits for each sample is indicated (n).**

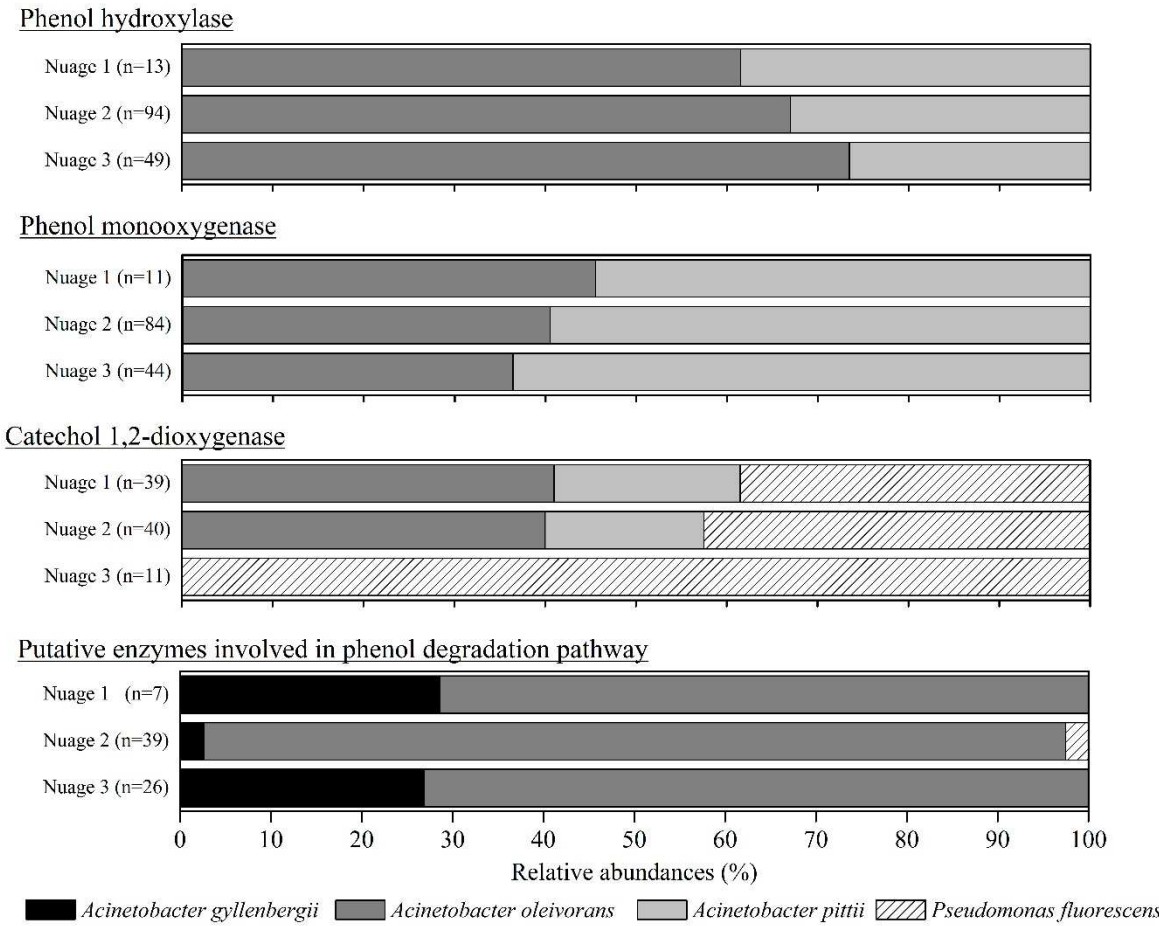

**Figure 3: Relative abundance of the putative taxonomic affiliation of the microorganisms involved in phenol degradation. For the 5 databases, microorganisms associated to a matching sequence with cloud transcripts are plotted here; the absolute total number of hits for each sample is indicated (n).**

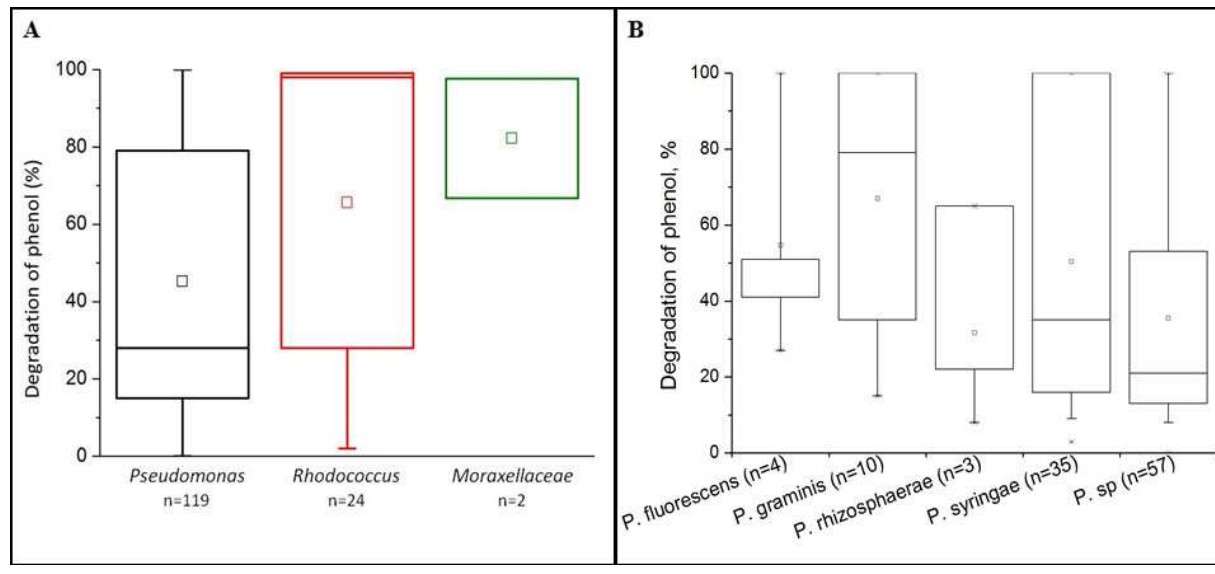


**Figure 4: Biodegradation of phenol by bacterial strains isolated from cloud waters. Results are expressed as the % of**
**phenol biodegradation measured by HPLC after 5 days of incubation at 17 °C. A) Results obtained for the 119**
***Pseudomonas* strains, the 24 *Rhodococcus* strains, and the 2 strains from the Moraxellaceae family. B) Focus on the**
**Pseudomonas species. Only species groups with a minimum of 3 strains are plotted here.**

**Table 1: Phenol concentrations measured by GC-MS in the five cloud water sampled at the PUY station.**

| Cloud water sampling date | Air mass origin | Phenol concentration ($\mu g\ L^{-1}$) |
|---|---|---|
| 05/11/2013[a] | West | 0.52 |
| 27/06/2014[a] | West | 0.73 |
| 16/02/216[a] | North East | 0.74 |
| 21/10/2016b | North West/North | 0.21 |
| 26/10/2016[b] | North West/North | 0.15 |

[a] From Lebedev et al. (2018), [b] This work.