# Peer review of "Potential for phenol biodegradation in cloud waters"

_Biogeosciences, 2018_

## Referee Comment (RC1) · O. BERGE (Referee) · 9 Jul 2018

General comments The study of the potential biodegradation of a major pollutant, phenol, by bacteria in cloud water presented in this paper is a pioneer work that is very important for understanding the global cycle of some toxic components and the activity of microorganisms in atmosphere. The use of both molecular and cuturable approaches is very convincing and these approaches are complementary. Metatranscriptomic analysis and biodegradation tests showed clearly the potentiality of phenol biodegradation in atmosphere and open question about the phenol biodegradation rates under realistic cloud conditions. The use of different cloud sampling for the different analyses (phenol quantification, metatranscriptomic analysis and phenol degradation tests) has to be better justified and taken into account in the discussion. Experimental design of the

biodegradation test has to be better explain. The distinction between the results from molecular versus culturable approaches need sometimes to be clarify. Major changes have to be done in the discussion about P. syringae strains.

Detailed comments Abstract : L 24 : Concentration of phenol in cloud samples was measured only in 2 samples in this work. The three other values are from a previous paper. Clarify. L 27 : Work has been done on strains not on isolates. Specify that the strains were isolated in a previous work. L 28 : Details on Puy station should be placed L 25 L 29-30 : Specify that the 3 samples were different of those used for phenol quantification L 35 : Specify that strains were selected in species known for having this activity

Introduction L 56 : Phenol and 4-ethylphenol are the most abundant phenols in clouds (Lebedev et al 2018, Table 2). Why have you limited the study to Phenol ? Are they degraded by the same enzymes? L 69-71 & 119-121 : It may be possible to decrease the reference number and keep the most significant L 123 : Add "in clouds" before the references L 126 : Metatranscriptomic allowed to detect gene expression (transcripts), it is more than simply detecting genes.

Materials & methods L 142-144 : 3 of the 5 samples of cloud water were extracted from Lebedev et al.2018 and only 2 were done in this study. Clarify. L 166-167 : Doing metatranscriptomic and phenol quantification on different samples must be better justify. Why choosing 3 consecutive periods of 5 h for the 3 samples for metatranscriptomic (are they considered as replicates ?), rather than 3 independent samples (different dates) ? It should be informative to have Hysplit information and phenol concentration for these 3 samples. L 169-172 : This control is great. Add some information on the transcriptomic result of it. L 188 : I think that it is Figure 1 that content information and not Figure 2. 189-190 : In these databases, have you included the "catechol operon" cited in Berge et al. 2014 that you compare to your data in the discussion (see details comments of the discussion below). Do these enzymes could be involved in other activities than phenol and catechol degradation? L 199-200 : Specify that strains

were isolated previously from different samples L 201 : The sampling of bacteria is not randomized why ? May be you should have found strains with degrading activity that you would'nt expected. The chosen strains expected to show an activity do not represent at all the cloud bacterial population. In the abstract you must explain that, before to give the percentage of positive strains for phenol biodegradation to not suggest that 93 % of cloud culturable bacteria are able to degrade phenol. L 201-207 : It should be great to know the abundance of these strains when isolated to have an idea of their importance in cloud (size of their population). L 204 : I expect that P. grimortii does not exist, check this name. Table SM1 : Specify that accession number is for 16S RNA gene sequence L 208 : In which volume were done the initial bacterial cultures ? L210 : Does Volvic water sterilized ? L212-218 : This section need to be better explained. It is not clear how bacterial concentrations were measured and when. Why have you chosen the x 104 factor ? 109 cells/ml seems to be very high bacterial concentration, justify. L 226-228 : Better to transfer this section in result or discussion: (it is already repeated L 311 -316) L 248-249 : What is the experimental design of this test ? Have you replicate the test ? If not, why ?

Result L 252 : previously isolated Table 1 : Usually, table have to be in column, with title in the first line. Unit could be in the title. L 265 : I think that you must cite Figure SM2 C, D, E in the text. L 266 : I should have write "Microbiote" L 272 :Table SM2, do not contain P. syringae sequences why (see related comments in the discussion part) ? L 275-277 : does this variability could be explain by various probabilities that a given degrading bacterial population encounter a given amount of phenol in cloud droplets ? This question has to be discussed somewhere. L 278-281 : Are these differences significant ? L283-285 : Is it the microbial activity or the microbial diversity that varied ? Is it in time or in space in the cloud ? This comment need to be clearer. L 286 : Figure 3 not Figure 2 L 290 : matching not matched. 2 sentences would be clearer. L292 : were tested ? were found in data bases L 288-299 : English has to be improved, to facilitate the understanding L295 : Clarify: which approach was used to calculate this 0.3 % ? Was it on the same samples studied in the metatranscriptomic analysis ?

L298 : Explain better why referring to Rhodococcus in this section. L 302 : culturable L 304-305 : Let us know how were selected the genus of interest ? From literature knowledge ? In comparison with table SM2 ? Anything else ? Figure 4 : why have you chosen to test many strains without replicate, when you may have chosen less strains with replication of the test? Actually, we have no idea of the test variability for one given strain. L 320 : "genus" not "strains" L 320 – 327 : Figure 4 B : Some degrading species of Pseudomonas are not present in databases used for bioinformatics. Genomes have been sequenced in some strains of these species, is it possible to find genes involved in phenol degradation in these genomes? L 325 : Reword this line : P. syringae is a species name not a genus. Which approach showed that P. syringae is the most abundant bacterial species in cloud water ?

Discussion & Conclusion: L 332 : Citation of Lebedev et al. 18 as a comparison is may not correct, when some data came from this paper L 333 : Add the range of phenol concentration found in all these papers (3.0 to 5.4 mgL-1 ?) L333-335 : Compare your data with those of Lebedev et al 2018. They found in the results section: "no major impact of the air mass origin" "The anthropization of the air masses seems to increase the levels of phenol and 4-nitrophenol in the clouds (our work and the literature)" To test the effect of air masses origins, you have two replicates of west origin and two from north west/north, which could be statistically compared. It will probably show, it is those from the non-polluted area that have the higher concentrations. Comment. L 335 : "Slight variation": I would say rather "in the same magnitude" because concentrations from West are approx. x 3 those of North west/North L 338: Not enzymes were detected, their trancripts were. L339: Sequences not species. Explain why referring here to Rhodococcus. You should say that you didn't find all the other bacterial species present in the data base (Table SM2) L 339: Replace "in parallel" by something like : "Culturable approach has shown previously than Rodoc and Pseudo were abundant but etc…(cite the papers)" L 343: Database constitution: what could you propose to improve the data base for phenol degradation ? You have tested strains of species that were not in data base and that showed phenol degradation. L 344: Culturable L
344: Aman et al. 1995 citation could be replaced if possible by a more recent one, containing estimations of percent of culturable bacteria done on cloud water or on substrates close to it. If you keep Aman et al. 95, percent of culturable bacteria in water were lower than 1 % if I well remember, check this. L 345-346: Better to say "strains from Acinetobacter, Pseudomonas and Rhodoccoccus genera are known to degrade phenol..." L 347 : Microbiote not microbiome. Specify if you speak about the microbiote described from the culturable or molecular approaches L 351: "Actinobacteria" is not useful here L 362-364: If possible cite only the main references reporting the range of values in surface water. L 365: Pseudomonas are more frequent in culturable bacteria of cloud water but not in metagenomic analysis. L 366-367: If P. syringae strains are able to degrade phenol (and may be other strains) why we don't find them in your data bases of phenol degradative enzymes (Table SM2) ? See related comments below on the ref : Bartoli et al. 2015. L 366-367 : Reformulate the sentence : not only P. syringae and P. graminis could be issued from phyllosphere. P. fluorescens, P. poae, P. viridiflava could also. L 369-382 and Table SM3 : Actually data from Berge et al 2014, are not pertinent for comparison with this study because the 763 strains studied in Berge et al. represented a very wide diversity (more than 20 potential species, see Gomila et al. 2017) when strains from clouds may represent less diversity. May be it would be possible to compare strains from clouds with those from other environments, clade by clade that would suppose to determine the exact phylogroup and clade classification of cloud strains. Concerning the catechol operon, Berge et al. said p 6: "Strains in this clade [clade 01b], as well as strains in phylogroup 3 [PG03], contain a catechol operon regrouping genes for degradation of aromatic compounds [32]". It means that only some strains and not all of them, had this catechol operon in their genome in these phylogroups. Therefore values of Table SM3 are note correct. The original data are reported in the ref 32 of the Berge et al (2014) paper. This ref (Bartoli et al 2015) will be more pertinent to cite in your study: it is shown that not all phylogroups were tested for the presence of catechol operon in their genome: only 19 strains from PG01 (14 positive), 4 strains from PG03 (3 positive) and one strain from

PG02 (negative). Therefore, Table SM3 and the related comments have to be profoundly modified. In particular, Bartoli et al. 2015 stated P 138 : "Comparison of gene content between publicly available genomes of several P. syringae pathogens of woody plants with those of herbaceous plants revealed an operon with predicted function in the catechol pathway that was present only in pathogens of woody plants". Again, why these sequences were not integrated in your database ? These authors also shown (p 137) that "All environmental strains [tested in the study] possessing an operon involved in the degradation of aromatic compounds via the catechol pathway grew endophytically and caused symptoms in kiwifruit vascular tissue". Concerning ice nucleating activity it is interesting to analyse the frequency of INA positive strains among the phenol degradative strains of P. syringae. Again it is not possible to compare with the percentages found by Berge et al. because of the very big difference in strain diversity but may be comparison of percentages clade by clade would be possible. You may discuss on the potential consequences of finding bacteria such as P. syringae, in clouds that have the catechol degrading operon linked with pathogenicity on woody plants, in terms of epidemiology, dispersion of pathogens and emergence of plant diseases. Discussion on the role of INA in this dispersion could be added. Discuss also the potential reverse consequences, the presence (manipulate or not) on phyllosphere of such P. syringae (or other phylospheric population) having the catechol degradative pathway operon, their driven by ascending air movements into the clouds and their effect on the phenol degradation in clouds (to be linked to comments on remediation L 424-425) L373 : In Berge et al. (2014) it is PG I & III that have catechol operon in their genomes and not PG I and II. Anyway these lines commenting Table SM3 have to be modified (see comments above) L 375 : Ice nucleation activity is not restricted to P. syringae and could be found mainly in Gammaproteobacteria, more specifically in Pseudomonadaceae (P. syringae, P. fluorescens), Xanthomonadaceae and Enterobacteriaceae. Why the study of INA was restricted to P. syringae ? L 389 : How can you assert that "enzymatic equipment for phenol degradation is largely present" ? Many strains were active, but they were chosen among the species assumed to be able to exhibit this

activity and there are no quantification of their abundance in cloud water, therefore, we have no idea of the real quantitative impact of these strains. L 412 focused L 416 why Rhodococcus when it was not found in the metatranscriptomic analysis ?

Technical comments Referring to previous work must be stated more clearly in the text. Words like, isolate, strain, species, genus have to be used in the good way. Italics for Latin names L 318 : two "."

---

## Referee Comment (RC2) · Anonymous Referee #2 · 13 Jul 2018

**REVIEW NOTE – bg-2018-251**

General comments

The Authors of the manuscript '*Potential for phenol biodegradation in cloud waters*' (bg-2018-251) isolated bacterial strains from cloud water (polluted with phenol) that are potentially capable of degrading phenol and its main degradation product (catechol). They also determined transcripts of genes coding for the enzymes responsible for phenol and catechol degradation including hydroxylase, monooxygenase and 1,2-dioxygenase. Based on these findings the Authors concluded that cloud water may be a potential environment for biotransformation of phenol by microorganisms including genus *Pseudomonas*, *Acinetobacter* and *Rhoddococcus*.

In my opinion, the study is interesting and has significant scientific value and novelty; however it needs some major revision. The methods have been properly designed and the results and reliable.

Specific comments

Major points:

In 'Introduction' some information concerning toxic effects of phenol (with appropriate references) must be provided because the statement that phenol is toxic is not satisfactory

In conclusion the statement: In conclusion, this is the first report of the potential degradation of phenol by cloud organism should be changed to (for example): In conclusion, this is the first report showing that cloud water is inhabited by microorganisms that have phenol degradation ability

Page 5, line 154-155, GC-MS analysis, how the samples were evaporated (what was the temperature during evaporation? or/and was nitrogen used to eliminate solvent?)

Page 8, Phenol HPLC analysis, which was the limit of detection and limit of quantification of phenol?

Minor points:

Abstract, line 17, correct 'particularly toxic' to 'toxic'
Abstract, line 25, please provide full name of 'PUY'
Abstract, line 25, correct '0.74 µg.L$^{-1}$' to '0.74 µg L$^{-1}$'
Introduction, line 47, correct 'high toxicity' to 'toxicity' (in fact phenol is less toxic than most of phenols of anthropogenic origin or/and numerous other xenobiotics)
Page 3, line 98, correct 'bacteria' to 'bacteria strains'
Page 4, line 107 and 113, correct 'opening' to 'cleavage'
Page 4, line 116, correct 'concentration' to 'density'
Page 6, line 187, correct 'proteins' to 'enzymes'
Page 8, line 239, correct to: '150 mm x 4.6 mm
Page 9, line 273 and 281, correct to: '2,3-dioxygenase
Page 9, line 281, correct 'opening' to 'cleavage'
Page 10, line 314, correct to: long induction periods of enzymes
Page 10, line 322-333, correct sentence
Page 11, line,338-341, correct sentence as it is not clear
Page 12, correct: 'surface water' to 'polluted surface water' as phenol does not occurs at high concentrations in natural surface waters
Page 12, line 374, please provide full name of 'INA+'

Page 12, line 375 and 382, correct Joly et al., 2013 and Berge et al., 2014 to Joly et al. (2013) and Berge et al. (2014)

Page 12, line 388-389, correct sentence

Page 13, line 395-396, correct 'shorter molecules' to e.g 'intermediates'

Page 13, line 396, correct 'opening' to 'cleavage'

Page 13, line 409, correct 'biological and abiotic' to 'biotic and abiotic'

Page 14, line 425, correct to (for example). 'The most probably, microorganisms could participate to phenol remediation in the atmosphere'

Figure 1, correct to: '1,2-dioxygenase' and '2,3-dioxygenase'

Figure 2, correct to: ' monooxygenase'

Figure 3, correct to: ' monooxygenase' and '1,2-dioxygenase'

Figure 4B, Y-axis, correct to: 'Degradation of phenol (%)' as it is in Figure 4A

Technical comments

In the whole manuscript, 'minutes' must be corrected to 'min', 'hours' to 'h'.

Please also write (for example) '25 ⁰C' instead of '25⁰C' and (for example) '1 mL' instead of '1mL'

English of the paper should be corrected in several places

---

## Author Comment (AC1) · 27 Jul 2018

First we would like to thank O. Berge for her interest in our work and her very detailed comments that will help to improve the manuscript; in particular the discussion about P. syringae strains will be deeply modified. The détails answers are presented as a Supplement PDF form (attached)

Please also note the supplement to this comment:
https://www.biogeosciences-discuss.net/bg-2018-251/bg-2018-251-AC1-supplement.pdf

**Supplement:**

**Answer to O. BERGE's comments**

First we would like to thank O. Berge for her interest in our work and her very detailed comments that will help to improve the manuscript; in particular the discussion about *P. syringae* strains will be deeply modified.

**Detailed comments**

**Abstract :**

Comment L 24 : Concentration of phenol in cloud samples was measured only in 2 samples in this work. The three other values are from a previous paper. Clarify not on isolates. Specify that

Answer: We agree, the text has been changed to:

" Phenol concentrations were measured by GC-MS on two cloud samples collected at the PUY station: they ranged from 0.15 to 0.21 µg.L$^{-1}$."

Comment L 27: Work has been done on strains the strains were isolated in a previous work.

Answer: The text has been changed to:

"From the 145 strains tested, 33 were isolated for this work."

Comment L 28 : Details on Puy station should be placed L 25

Answer: Done

Comment L 29-30 : Specify that the 3 samples were different of those used for phenol quantification

Answer: Done

Comment L 35 : Specify that strains were selected in species known for having this activity

Answer: Done

**Introduction**:

Comment  L 56 : Phenol and 4-ethylphenol are the most abundant phenols in clouds (Lebedev et al 2018, Table 2). Why have you limited the study to Phenol ? Are they degraded by the same enzymes?

Answer:  We choose to focus on phenol rather than on 4-ethylphenol for two reasons: first of all when we started this project we only had the measurements of Phenol in clouds, both in our samples and also in the literature; second  more data are available in the literature on the aerobic biodegradation of phenol. The biodegradation pathways of these two compounds are very different, referring to KEGG Bisphenol degradation pathway, 4-ethylphenol is oxidized to 1-(4'-hydroxyphenyl) ethanol and ends up in Hydroquinone which then enters the chlorocyclohexane and chlorobenzene degradation pathway.

Comment L 69-71 & 119-121 : It may be possible to decrease the reference number and keep the most significant

Answer: done

L 123 : Add "in clouds" before the references

Answer: done

Comment L 126 : Metatranscriptomic allowed to detect gene expression (transcripts),it is more than simply detecting genes

Answer: the text has been changed

**Materials & methods:**

Comment L 142-144 : 3 of the 5 samples of cloud water were extracted from Lebedev et al.2018 and only 2 were done in this study. Clarify.

Answer: The text has been changed to "Two cloud water samples collected in 2016 (October 21$^{th}$ and October 26$^{th}$) were analyzed in this work for phenol quantification by GC-MS. Three other samples were previously sampled and analyzed in 2013 (November 05$^{th}$), 2014 (June 27$^{th}$) and 2016 (February 16$^{th}$) (Lebedev et al., 2018)."

Comment L 166-167 : Doing metatranscriptomic and phenol quantification on different samples must be better justify. Why choosing 3 consecutive periods of 5 h for the 3 samples for metatranscriptomic (are they considered as replicates ?), rather than 3 independent samples (different dates) ? It should be informative to have Hysplit information and phenol concentration for these 3 samples.

Answer: Cloud samples are difficult to collect, so a limited amount of sample is available and it is not possible to carry different types of experiments at the same time as experimental conditions for GC-MS analysis and metatranscriptomics are very different. For metatranscriptomics RNA later solution is added directly in the cloud collectors, this is incompatible for other chemical analysis. For GC-MS special care has to be taken to avoid all chemical contaminations as we are measuring traces of compounds.

These 3 samples are indeed technically not true replicates, but these were collected from a single cloud event which remained stable over time, in terms of meteorology (see air mass backward trajectories in SM) and chemical content (not shown as this is currently under review elsewhere; we can provide the data if asked).

This information has been included in the text in 2.3 .

"The cloud event remained stable over the duration of sampling as attested by air mass backward trajectories (Figure SM 2). "

[Figure]

Figure SM2: Back trajectories of the air masses corresponding to cloud events at the PUY station. 24 h back-trajectories of the air masses sampled were determined using the HYSPLIT model (HYbrid Single-Particle Lagrangian Integrated Trajectory). Cloud water samples were collected on November 17th, 2014, for three consecutive periods of 5 hours.

Comment L 169-172 : This control is great. Add some information on the transcriptomic result of it.

Answer:

More information about the control has been added in section 2.3.

"These controls were clearly distinct from samples: based on their contribution to identified ribosome sequences, these contained mostly Enterobacteriaceae (66%), Dikarya (9.2%), Streptococcaceae (5.4%), Vibrionaceae (2.8%) and Micrococcaceae (1.2%), i.e. not the taxa of interest here. Conservatively, the sequences present in controls were further removed from sample files (BWA-MEM; li et al., 2013). "

Li H. Aligning sequence reads, clone sequences and assembly contigs with BWA-MEM. *arXiv:13033997 [q-bio]* 2013.

Comment L 188 : I think that it is Figure 1 that content information and not Figure 2.

Answer: thank you, this was a mistake (corrected)

Comment L 189-190 : In these databases, have you included the "catechol operon" cited in Berge et al. 2014 that you compare to your data in the discussion (see details comments of the discussion below). Do these enzymes could be involved in other activities than phenol and catechol degradation?

Answer: We agree that this would have been very interesting. However we did not include them because the sequences for phenol biodegradation were only searched in NCBI data bank while most

of the operon catechol sequences from Berge et al. were deposited in PAMDG.org. In addition in NCBI the only sequences belonging to *Pseudomonas syringae* were described as coding for "hypothetical proteins" and were not thus considered in our list as we excluded all "hypothetical sequences ".

Comment L 199-200 : Specify that strains were isolated previously from different samples

Answer: we added this sentence:

From the 145 strains tested, 33 of the strains were isolated for this work, the others were published earlier (see Table SM1).

Comment L 201: The sampling of bacteria is not randomized why? May be you should have found strains with degrading activity that you would'nt expected. The chosen strains expected to show an activity do not represent at all the cloud bacterial population. In the abstract you must explain that, before to give the percentage of positive strains for phenol biodegradation to not suggest that 93 % of cloud culturable bacteria are able to degrade phenol.

Answer:  We choose to test specifically *Pseudomonas and Acinteobacter* strains as they were detected in our metatranscriptomic analysis. In addition *Rhodococcus* is also well-known to biodegrade phenol in the literature (as well as *Pseudomonas* and *Acinetobacter*). As no *Acinetobacter* was available in our bacterial collection we choose closely related genera namely two strains of *Moraxella* and *Psychrobacter*.  In addition *Pseudomonas* and *Rhodococcus* are most frequently genera found in culturable bacteria from clouds (see Renard et al., Atmos. Chem. Phys. 2016; Vaitilingom et al, Atmos. Environ. 2012). We preferred to choose potentially positive strains which numbers are representative of the cloud microbiota  instead of looking at random because in many cases strains are in a very limited number when the genus is considered.

In the abstract we did wrote "Bacterial isolates from cloud water samples (*Pseudomonas* spp*., Rhodococcus*  spp. and strains from the Moraxellaceae family were screened for their ability to degrade phenol: 93% of the 145 strains tested were positive"… which is clearly different from " 93% the cloud culturable bacteria".

Comment L 201-207 : It should be great to know the abundance of these strains when isolated to have an idea of their importance in cloud (size of their population).

Answer:

This information has been added to the text.

 "*Pseudomonas*  and *Rhodococcus* strains represent 20.4 % and 4.10% of the 584 strains of our cloud bacterial collection. From our experience, at the genus level, *Pseudomonas* and *Rhodococcus* are among the most frequent bacteria in clouds: *Pseudomonas* strains in particular have been frequently isolated by culture (Vaïtilingom et al., 2012; Joly et al., 2013), and both targeted and untargeted molecular analyses (and metagenomes, respectively) demonstrated high occurrence in the bacterial communities. These represented 0.1 to >2% of the prokaryotes ribosome sequences in amplicon sequencing investigations (Amato et al., 2017). Based on the biomass in clouds (~$10^4$ bacteria cells/mL; Vaïtilingom et al., 2012), and assuming even ribosome amplification between bacterial groups, we can infer the presence of ~$10^3$ *Pseudomonas*/mL and ~$10^2$ *Rhodococcus*/mL of cloud water."

Comment L 204 : I expect that *P. grimortii* does not exist, check this name. Table SM1 : Specify that accession number is for 16S RNA gene sequence

Answer: This was a tipping error the exact name is *Pseudomonas grimontii* . Table SM1 was modified.

Comment L 208 : In which volume were done the initial bacterial cultures ?Does Volvic water sterilized ?

Answer: the volume was 25mL of R2A media, Volvic® water was previously sterilized by filtration under sterile conditions using a 0.22µm PES filter.

This information was added to the revised manuscript.

Comment L212-218 : This section need to be better explained .It is not clear how bacterial concentrations were measured and when. Why have you chosen the x $10^4$ factor ? $10^9$ cells/ml seems to be very high bacterial concentration, justify.

Answer: we agree that we need to better explain this section. Rough bacterial concentration was estimated from the OD measurement during the growth and finally confirmed by Flow cytometry. The concentration of phenol (100 µM) was chosen to be measured easily by UV-HPLC. This value being fixed, we have adapted the cell concentration to keep the ratio phenol/cells found in cloud waters (0.008µM of phenol (0.74 µg/ L) and $10^5$ cells/ mL in cloud sample is equivalent to 80 µM of phenol and $10^9$ cells/mL in our experiment). As explained in this section we estimate that with a constant ratio the rates of degradation are comparable.

Comment L 226-228 : Better to transfer this section in result or discussion: (it is already repeated L 311 -316)

Answer: done

Comment L 248-249 : What is the experimental design of this test ? Have you replicate the test ? If not, why ?

Answer: No replicates were performed as screening 145 strains represents a lot of work and time. We made 3 replicates only for one strain (see later we showed a high reproducibility as the rate of biodegradation was 4.70 ± 2.06  $10^{-17}$mole. cell$^{-1}$.h$^{-1}$). The idea was to test a maximum of strains. The final result being a % of degradation after 5 days and not a very precise rate of degradation for all these strains.

We used a non-parametrical Kruskal–Wallis analysis because the distribution of our data is not a normal one, this is due to the large difference in the number of strains in each category.

**Results:**

Comment L 252 : previously isolated

Answer: From  the 145 strains tested, 33 were isolated for this work. The text has been changed accordingly.

Comment Table 1 : Usually, table have to be in column, with title in the first line. Unit could be in the title. L 265 : I think that you must cite Figure SM2 C, D, E in the text. L 266 : I should have write "Microbiote"

Answer: OK

**Table 1:** Phenol concentrations measured by GC-MS in the five cloud water sampled at the PUY station.

| Cloud water sampling date | Air mass origin | Phenol concentration |
|---|---|---|
| 05/11/2013[a] | West | 0.52 µg.L$^{-1}$ |
| 27/06/2014[a] | West | 0.73 µg.L$^{-1}$ |
| 16/02/216[a] | North East | 0.74 µg.L$^{-1}$ |
| 21/10/2016b | North West/North | 0.21 µg.L$^{-1}$ |
| 26/10/2016[b] | North West/North | 0.15 µg.L$^{-1}$ |

[a]From Lebedev et al. (2018), [b]This work.

Comment L 272 :Table SM2, do not contain *P. syringae* sequences why (see related comments in the discussion part)?
Answer: We agree that this would have been very interesting. However we did not include them because these sequences did not came out during our search in NCBI data bank
Comment L 275-277 : does this variability could be explain by various probabilities that a given degrading bacterial population encounter a given amount of phenol in cloud droplets? This question has to be discussed somewhere.
Answer: This slight variability is due to the changes in the biodiversity present in each sample ( Unpublished data) but is not really significant (see bellow) , the text has been changed as follows:

"However the slight differences observed between the 3 cloud samples are not significant when analyzed by a non-parametrical Kruskal-Wallis test."

Comment L 278-281: Are these differences significant ?
Answer: As suggested by the referee we analyzed our data with a Kruskal-Wallis test, the result shows that the differences between the 3 cloud water samples (Figures 2 and 3) are not significant.
Comment L283-285 : Is it the microbial activity or the microbial diversity that varied ?Is it in time or in space in the cloud ? This comment need to be clearer.
 Answer: The 3 cloud samples are not statistically different..
Comment L 286 : Figure3 not Figure 2 L 290 : matching not matched.
Answer: We think that "matched" is correct
Comment :2 sentences would be clearer. L292: were tested ? were found in data bases L 288-299 : English has to be improved to facilitate the understanding
Answer: "tested" was replaced by "used for our search in data bases"

Comment L295 : Clarify: which approach was used to calculate this 0.3 % ? Was it on the same samples studied in the metatranscriptomic analysis ?

Answer:

We acknowledge that the way these values were reported was misleading. The study has now been published (Amato et al., 2017a) and the text has been modified accordingly with our final analysis.

"Gamma-proteobacteria were found to contribute up to 21% of the ribosome sequences identified in bacteria in targeted sequencing investigations. *Pseudomonas* in particular was highlighted as one of the most represented genus (contributing alone up to 2% of the ribosome sequences) and most active genera based on its representation in transcriptomes and consecutive high ribosomal cDNA:DNA ratio (Amato et al, 2017a, Figure SM4). *Acinetobacter* and *Rhodococcus* were much less represented (<0.1% of the ribosome sequences) but also accounted for groups of interest regarding potential metabolic activity."

Comment L298 : Explain better why referring to *Rhodococcus* in this section. L 302 : culturable
Comment L 304-305 : Let us know how were selected the genus of interest ? From literature knowledge ? In comparison with table SM2 ? Anything else ?

Answer: We have change the text to :

"We choose to test specifically *Pseudomonas* and *Acinteobacter* strains as they were detected in our metatranscriptomic analysis. As no *Acinetobacter* was available in our bacterial collection we choose closely related genera namely two strains of *Moraxella* and *Psychrobacter*. In addition *Rhodococcus* is well-known to biodegrade phenol in the literature (as well as *Pseudomonas* and *Acinetobacter*). *Pseudomonas* and *Rhodococcus* are also the most frequently found genera in culturable bacteria from clouds (Renard et al., 2016; Vaitilingom et al., 2012)."

Comment Figure 4 : why have you chosen to test many strains without replicate, when you may have chosen less strains with replication of the test? Actually, we have no idea of the test variability for one given strain.

Answer: the important point here was to test the maximum number of strains chosen on the criteria indicated above. Our goal was not in this paper to give very accurate biodegradation rates but to estimate the potential of biodegradation of the strains. Screening 145 strains represents a lot of experiment work. We did calculate the standard deviation of phenol biodegradation rates by one of the strains (unpublished) with 3 replicates (3 independent experiments with different cultures) and we obtained this value: $4.70 \pm 2.06 \ 10^{-17}$mole. cell$^{-1}$.h$^{-1}$, it shows that the experiments are highly reproducible.

Comment L 320 : "genus" not "strains"

Answer: We used "strains" and not "genus" because in the same sentence we have *Pseudomonas* and *Rhodococcus* which are genera but also *Moraxellaceae* which is not a genus but a family.

Comment L 320 – 327 Figure 4 B Some degrading species of *Pseudomonas* are not present in databases used for bioinformatics. Genomes have been sequenced in some strains of these species, is it possible to find genes involved in phenol degradation in these genomes?

Answer: We did not look for that so we do not have any information.

Comment L 325 : Reword this line : *P. syringae* is a species name not a genus.

Answer: done

**Comment** Which approach showed that *P. syringae* is the most abundant bacterial species in cloud water ?

Answer: This is based on the cultural approach (See Renard et al 2016). This information has been added in the text.

**Discussion & Conclusion:**

Comment L 332 : Citation of Lebedev et al. 18 as a comparison is may not correct, when some data came from this paper L 333 : Add the range of phenol concentration found in all these papers (3.0 to 5.4 $\mu gL^{-1}$ ?)

Answer: we agree, the reference Lebedev et al. 2018 was deleted and we add the range of phenol concentration found in all these papers (3.0 to 5.4 $\mu gL^{-1}$)

Comment L333-335 : Compare your data with those of Lebedev et al 2018. They found in the results section: "no major impact of the air mass origin" "The anthropization of the air masses seems to increase the levels of phenol and 4-nitrophenol in the clouds (our work and the literature)" To test the effect of air masses origins, you have two replicates of west origin and two from north west/north, which could be statistically compared. It will probably show, it is those from the non-polluted area that have the higher concentrations.

"Slight variation": I would say rather "in the same magnitude" because concentrations from West are approx. x 3 those of North west/North

Answer: We changed the text as follows:

"Although the concentration of phenol remains within the same order of magnitude in the 5 cloud samples, it seems that the origin of the air masses had an impact on this concentration, it was 3 times lower in non-polluted air masses (west) than in polluted ones (north west/north)."

Comment L 338: Not enzymes were detected, their trancripts were.

Sequences not species. Explain why referring here to *Rhodococcus*. You should say that you didn't find all the other bacterial species present in the data base (Table SM2)

Answer: We changed the text as follows: "…transcripts belonging to *Pseudomonas* and *Acinetobacter* strains but not for the sequences of the other strains present in Table SM2. It was surprising not to find *Rhodococcus* sequences as this genus is well-known to degrade phenol in the literature. "

Comment L 339: Replace "in parallel" by something like: "Culturable approach has shown previously than Rodoc and Pseudo were abundant but etc: : :(cite the papers)"

Answer: see change above

Comment L 343: Database constitution: what could you propose to improve the data base for phenol degradation ? You have tested strains of species that were not in data base and that showed phenol degradation.

Answer: We have added this sentence.

"In the future the database for phenol degradation could be improved by integrating more sequences, especially considering other data banks than NCBI. For instance the Catechol operon sequences of *Pseudomonas synringae* (Berge et al 2014) could be added to the data base. We recently published the genome sequence of *Pseudomonas syringae* 32b-74, *Pseudomonas graminis* 13b-3 and *Rhodococcus enclensis* 23b-28 which are degrading phenol (Table 1)(Besaury et al 2017a, b and Lallement et al 2017); they could be used to implement the database. Finally in the future the genome of many phenol degraders (table 1) could be also sequenced and integrated."

L. BESAURY, P. AMATO, N. WIRGOT, M. SANCELME, A.-M. DELORT. Draft genome sequence of *Pseudomonas graminis* PDD-13b-3, a model strain isolated from cloud water. *Genome Announcement*, 2017a, **5**:e00464-17.

L. BESAURY, P. AMATO, M. SANCELME, A.-M. DELORT. Draft genome sequence of *Pseudomonas syringae* PDD-32b-74, a model strain for ice nucleation studies in the atmosphere. *Genome Announcement*, 2017b, **5**:e00742-17.

A. LALLEMENT, L. BESAURY, B. EYHERAGUIBEL, P. AMATO, M. SANCELME, G. MAILHOT, A.-M. DELORT. Draft genome sequence of *Rhodococcus enclensis* PDD-23b-28, a model strain isolated from cloud water. *Genome Announcement*, 2017, **5**:e01199-17.

Comment L 344: Culturable

Answer: done

Comment  L 344: Aman et al. 1995 citation could be replaced if possible by a more recent one, containing estimations of percent of culturable bacteria done on cloud water or on substrates close to it. If you keep Aman et al. 95, percent of culturable bacteria in water were lower than 1 % if I well remember, check this.

Answer:

In order to be more specific, the reference Amann et al .reference has been completed by our study on clouds (Vaïtilingom et al. 2012).

Comment  L 345-346: Better to say "strains from *Acinetobacter*, *Pseudomona*s and *Rhodoccoccus* genera are known to degrade phenol: : :"

Answer: Changed

Comment  L 347 : Microbiote not microbiome. Specify if you speak about the microbiote described from the culturable or molecular approaches

Answer: Changed to "The cloud microbiota as described from a culturable approach…"

Comment L 351: "Actinobacteria"is not useful here

Answer: deleted

Comment L 362-364: If possible cite only the main references reporting the range of values in surface water.

Answer: Done

Comment L 365: *Pseudomonas* are more frequent in culturable bacteria of cloud water but not in metagenomic analysis.

Answer: *Pseudomonas* are more frequent in culturable bacteria of cloud water but in addition they belong to the most active strains as shown by metatrancriptomic's data (Amato et al, PlosONE 2017) We changed the text as follows:

"In our study we focused on *Pseudomonas* strains as they are the most frequent culturable strains (Vaïtilingom et al., 2012) and belong to the most active strains in cloud waters (Amato et al., 2017a)."

Comment  L 366-367: If *P. syringae* strains are able to degrade phenol (and may be other strains) why we don't find them in your data bases of phenol degradative enzymes (Table SM2) ? See related comments below on the ref : Bartoli et al. 2015.

Answer: As explained earlier the sequences were not integrated as there were not in NCBI data bank. They could be integrated in the future.

Comment  L 366-367 : Reformulate the sentence : not only *P. syringae* and *P. graminis* could be issued from phyllosphere. *P. fluorescens*, *P. poae*, *P. viridiflava* could also.

Answer: *P. fluorescens*, *P. poae* and *P. viridiflava* were added to the list.

Comment L 369-382 and Table SM3 : Actually data from Berge et al 2014, are not pertinent for comparison with this study because the 763 strains studied in Berge et al. represented a very wide diversity (more than 20 potential species,  see Gomila et al. 2017) when strains from clouds may

represent less diversity. May be it would be possible to compare strains from clouds with those from other environments, clade by clade that would supposed to determine the exact phylogroup and clade classification of cloud strains. Concerning the catechol operon, Berge et al. said p 6:"Strains in this clade [clade 01b], as well as strains in phylogroup 3 [PG03], contain a catechol operon regrouping genes for degradation of aromatic compounds [32]". It means that only some strains and not all of them, had this catechol operon in their genome in these phylogroups. Therefore values of Table SM3 are note correct. The original data are reported in the ref 32 of the Berge et al (2014) paper. This ref (Bartoli et al 2015) will be more pertinent to cite in your study: it is shown that not all phylogroups were tested for the presence of catechol operon in their genome: only 19 strains from PG01 (14 positive), 4 strains from PG03 (3 positive) and one strain from C5 PG02 (negative). Therefore, Table SM3 and the related comments have to be profoundly modified.

Answser: Thank you for clarifying this point, indeed we understood that all the strains (and not some strains) from phylogroups 1 and 3 contained the catechol operon. So it is clear that Table SM3 and all the related comments in the discussion should be deleted.

Comment :In particular, Bartoli et al. 2015 stated P 138 : "Comparison of gene content between publicly available genomes of several *P. syringae* pathogens of woody plants with those of herbaceous plants revealed an operon with predicted function in the catechol pathway that was present only in pathogens of woody plants". Again, why these sequences were not integrated in your database ? These authors also shown (p 137) that "All environmental strains [tested in the study] possessing an operon involved in the degradation of aromatic compounds via the catechol pathway grew endophytically and caused symptoms in kiwifruit vascular tissue". Concerning ice nucleating activity it is interesting to analyze the frequency of INA positive strains among the phenol degradative strains of *P. syringae*. Again it is not possible to compare with the percentages found by Berge et al. because of the very big difference in strain diversity but may be comparison of percentages clade by clade would be possible. You may discuss on the potential consequences of finding bacteria such as *P. syringae*, in clouds that have the catechol degrading operon linked with pathogenicity on woody plants, in terms of epidemiology, dispersion of pathogens and emergence of plant diseases. Discussion on the role of INA in this dispersion could be added. Discuss also the potential reverse consequences, the presence (manipulate or not) on phyllosphere of such *P. syringae* (or other phylospheric population) having the catechol degradative pathway operon, their driven by ascending air movements into the clouds and their effect on the phenol degradation in clouds (to be linked to comments on remediation L 424-425)

Answer: The reviewer's comment is extremely useful and opens new points to discuss in our paper. We added this text:

"Interestingly Bartoli et al (2015) showed that the genome of several *P. syringae* pathogens of woody plants contained a catechol operon, while it was not the case for other *P. syringae* strains pathogens of herbaceous plants. These results strongly suggested that the presence of enzymes present in the catechol pathway could help the degradation of aromatics present in lignins. In addition Berge et al. (2014) showed that some *P. syringae* strains from phylogroups 1 and 3 that were Ice Nuclei Active (INA[+]) also contained the catechol operon. In our case we also measured the ice nucleation activity of the 35 *Pseudomonas syringae* strains as described in Joly et al. (2013). Figure SM5 presents the strains which were INA+ (T>-8°C) *versus* their phenol degradation ability. Among the phenol degrader, 57.6% of the bacteria were INA[+].

Clouds can be considered as a long-distance shuttle for microorganism transport and INA$^+$ bacteria are suspected to induce precipitations and thus participate to the water cycle (Morris et al., 2008). Consequently, the presence of *Pseudomonas syringae* in clouds combining ice nucleation and phenol degradation properties can be of major importance for the pathogenicity on woody plants, in terms of epidemiology, dispersion of pathogens and emergence of plant diseases."

Bartoli, C., Lamichane, J.R., Berge, O., Guilbaud, C., Varvaro, L., Balestra, G.M., Vinatzer, B.A. and Morris, C.: A framework to gauge the epidemic potential of plant pathogens in environmental reservoirs: the example of kiwifruit canker. Mol. Plant  Pathol. 16, 137-149, 2015.  DOI: 10.1111/mpp.12167.
Morris, C.E., Sands, D. C.,Vinatzer, B.A., Glaux, C., Guilbaud, C., Buffière, A., Yan, S., Dominguez, H. and Thompson, B.M. :The life history of the plant pathogen *Pseudomonas syringae* is linked to the water cycle *. ISME J.,* 2, 321–334, 2008.

Comment L373 : In Berge et al. (2014) it is PG I & III that have catechol operon in their genomes and not PG I and II. Anyway these lines commenting Table SM3 have to be modified (see comments above)
Answer: This part has been deleted
Comment L 375 : Ice nucleation activity is not restricted to *P. syringae* and could be found mainly in Gammaproteobacteria, more specifically in Pseudomonadaceae (*P. syringae*, *P. fluorescens*), Xanthomonadaceae and Enterobacteriaceae. Why the study of INA was restricted to *P. syringae* ?
Answer: Initially these results were reported to be compared with those of Berge et al (2014) connecting IN and Catechol operons. This is why IN results were limited to *P. syringae*. It could be interesting in the future to test other members of our collection for IN activity.
 Comment L 389 : How can you assert that "enzymatic equipment for phenol degradation is largely present" ? Many strains were active, but they were chosen among the species assumed to be able to exhibit this activity and there is no quantification of their abundance in cloud water, therefore, we have no idea of the real quantitative impact of these strains.
Answer: We modified the text as follows:
" On the other hand the large screening performed with selected cloud strains showed that they have the enzymatic equipment for phenol degradation. Future work should be conducted to evaluate this potential for phenol biodegradation in real clouds where a larger microbial diversity is present. In particular, precise biodegradation rates should be  determined under realistic cloud conditions to evaluate its real impact. It will be also very important…"

Comment L 412 focused
Answer: Changed
Comment L 416 : why *Rhodococcus* when it was not found in the metatranscriptomic analysis ?
Answer: Actually this sentence refers both to the metranscriptomic analysis and the screening. We added :
"These two combined approaches suggested that *Pseudomonas*, *Acinetobacter* and *Rhodococcus*…."
Technical comments Referring to previous work must be stated more clearly in the text.
Words like, isolate, strain, species, genus have to be used in the good way. Italics for
Latin names L 318 : two "."
Done

---

## Author Comment (AC2) · 27 Jul 2018

**REVIEW NOTE – bg-2018-251**

First of all we would like to thank the reviewer for his/her constructive comments on our paper.

**General comments**
The Authors of the manuscript '*Potential for phenol biodegradation in cloud waters*' (bg-2018-251) isolated bacterial strains from cloud water (polluted with phenol) that are potentially capable of degrading phenol and its main degradation product (catechol). They also determined transcripts of genes coding for the enzymes responsible for phenol and catechol degradation including hydroxylase, monooxygenase and 1,2-dioxygenase. Based on these findings the Authors concluded that cloud water may be a potential environment for biotransformation of phenol by microorganisms including genus *Pseudomonas*, *Acinetobacter* and *Rhodococcus*.

In my opinion, the study is interesting and has significant scientific value and novelty; however it needs some major revision. The methods have been properly designed and the results and reliable.

Specific comments

**Major points:**
Comment: In 'Introduction' some information concerning toxic effects of phenol (with appropriate references) must be provided because the statement that phenol is toxic is not satisfactory
Answer: This sentence has been added in the revised version with some references.
Phenol has an environmental impact, particularly on the aquatic biota (microorganisms, protozoa, invertebrates, and vertebrates) (Babich and Davis, 1981, Duana et al., 2018). Phenol represents also a risk for human beings because it can be rapidly absorbed through the skin and by inhalation through the lungs. In particular it provokes cutaneous exfoliation and cardiac arrhythmias; it is also toxic to the liver and kidneys (Babich and Davis, 1981; Lober , 1987). For more information the National Library of Medicine HSDB Database  can be searched. .

Babich, H. and Davis, D.L.: Phenol: A review of environmental and health risks, Regulatory Toxicology and Pharmacology, 1, 90-109, 1981.

Duana, W., Menga, F., Cuia, H., Linc, Y., Wangc, G., and Wuc J.: Ecotoxicity of phenol and cresols to aquatic organisms: A review, Ecotoxicol. Environ. Safety 157, 441–456, 2018.https://doi.org/10.1016/j.ecoenv.2018.03.089.

Lober, C.W.: Chemexfoliation--indications and cautions. J. Am. Acad. Dermatol. 17, 109-112, 1987.

National Library of Medicine HSDB Database -PHENOL -
*https://toxnet.nlm.nih.gov/cgi-bin/sis/search/a?dbs+hsdb:@term+@DOCNO+113.*

Comment: In conclusion the statement: In conclusion, this is the first report of the potential degradation of phenol by cloud organism should be changed to (for example): In conclusion, this is the first report showing that cloud water is inhabited by microorganisms that have phenol degradation ability
Answer: the text has been changed as requested.
Comment: Page 5, line 154-155, GC-MS analysis, how the samples were evaporated (what was the temperature during evaporation? or/and was nitrogen used to eliminate solvent?)

Answer: We added. "...evaporated to 1mL using a rotary evaporator under reduced pressure; temperature of the water bath was 20°C."

Comment: Page 8, Phenol HPLC analysis, which was the limit of detection and limit of quantification of phenol?

Answer: As stated in the Material and Method section "Limit of phenol quantification was 3.8 µM. Strains are not considered active below 5 % of phenol degradation, corresponding to 5 µM."

**Minor points**:

All requested corrections have been made in the revised version

Abstract, line 17, correct 'particularly toxic' to 'toxic'

Abstract, line 25, please provide full name of 'PUY'

Abstract, line 25, correct '0.74 µg.L$_{-1}$' to '0.74 µg L$_{-1}$'

Introduction, line 47, correct 'high toxicity' to 'toxicity' (in fact phenol is less toxic than most of phenols of anthropogenic origin or/and numerous other xenobiotics)

Page 3, line 98, correct 'bacteria' to 'bacteria strains'

Page 4, line 107 and 113, correct 'opening' to 'cleavage'

Page 4, line 116, correct 'concentration' to 'density'

Page 6, line 187, correct 'proteins' to 'enzymes'

Page 8, line 239, correct to: '150 mm x 4.6 mm

Page 9, line 273 and 281, correct to: '2,3-dioxygenase

Page 9, line 281, correct 'opening' to 'cleavage'

Page 10, line 314, correct to: long induction periods of enzymes

Page 10, line 322-333, correct sentence

Page 11, line,338-341, correct sentence as it is not clear

Page 12, correct: 'surface water' to 'polluted surface water' as phenol does not occurs at high concentrations in natural surface waters

Comment: Page 12, line 374, please provide full name of 'INA+'

Answer: "Ice Nuclei Active"

Page 12, line 375 and 382, correct Joly et al., 2013 and Berge et al., 2014 to Joly et al. (2013) and Berge et al. (2014)

Page 12, line 388-389, correct sentence

Page 13, line 395-396, correct 'shorter molecules' to e.g 'intermediates'

Page 13, line 396, correct 'opening' to 'cleavage'

Page 13, line 409, correct 'biological and abiotic' to 'biotic and abiotic'

Page 14, line 425, correct to (for example). 'The most probably, microorganisms could participate to phenol remediation in the atmosphere'

Figure 1, correct to: '1,2-dioxygenase' and '2,3-dioxygenase'

Figure 2, correct to: ' monooxygenase'

Figure 3, correct to: ' monooxygenase' and '1,2-dioxygenase'

Figure 4B, Y-axis, correct to: 'Degradation of phenol (%)' as it is in Figure 4A

Technical comments

In the whole manuscript, 'minutes' must be corrected to 'min', 'hours' to 'h'.

Please also write (for example) '25 °C' instead of '25°C' and (for example) '1 mL' instead of '1mL'

English of the paper should be corrected in several places